

# The proportion of core species in a community varies with spatial scale and environmental heterogeneity

Molly F. Jenkins[1], Ethan P. White[2,3,4] and Allen H. Hurlbert[1,5]

[1] Environment, Ecology, and Energy Program, University of North Carolina at Chapel Hill, Chapel Hill, NC, United States of America
[2] Department of Wildlife Ecology and Conservation, University of Florida, Gainesville, FL, United States of America
[3] Informatics Institute, University of Florida, Gainesville, FL, United States of America
[4] Biodiversity Institute, University of Florida, Gainesville, FL, United States of America
[5] Department of Biology, University of North Carolina at Chapel Hill, Chapel Hill, NC, United States of America

Corresponding author
Allen H. Hurlbert,
Hurlbert@bio.unc.edu

## ABSTRACT

Ecological communities are composed of a combination of core species that maintain local viable populations and transient species that occur infrequently due to dispersal from surrounding regions. Preliminary work indicates that while core and transient species are both commonly observed in community surveys of a wide range of taxonomic groups, their relative prevalence varies substantially from one community to another depending upon the spatial scale at which the community was characterized and its environmental context. We used a geographically extensive dataset of 968 bird community time series to quantitatively describe how the proportion of core species in a community varies with spatial scale and environmental heterogeneity. We found that the proportion of core species in an assemblage increased with spatial scale in a positive decelerating fashion with a concomitant decrease in the proportion of transient species. Variation in the shape of this scaling relationship between sites was related to regional environmental heterogeneity, with lower proportions of core species at a given scale associated with high environmental heterogeneity. Understanding this influence of scale and environmental heterogeneity on the proportion of core species may help resolve discrepancies between studies of biotic interactions, resource availability, and mass effects conducted at different scales, because the importance of these and other ecological processes are expected to differ substantially between core and transient species.

# INTRODUCTION

Species differ in the temporal persistence with which they occur at any given site. While some species are reliably observed year in and year out, others appear only occasionally (*Ulrich & Ollik, 2004*; *Belmaker, 2009*; *Dolan et al., 2009*; *Gaston et al., 2007*; *Umaña et al., 2017*). Indeed, recent work from a broad range of ecological communities has shown that temporal occupancy is typically bimodal, reflecting these two groups which have been

referred to as "core" and "transient" species (*Coyle, Hurlbert & White, 2013*; *Umaña et al., 2017*; *Taylor et al., 2018*). Core species, in persisting at a site over time, are thought to maintain viable populations through successful reproduction (*Coyle, Hurlbert & White, 2013*; *Taylor et al., 2018*). In contrast, transients do not persist reliably, and presumably do not maintain viable populations (*Magurran & Henderson, 2003*; *Umaña et al., 2017*). Ecologists have typically ignored this distinction and have assumed that the complete list of species observed over some biological survey constitutes a meaningful "community" of interest for analysis. However, core and transient species interact with their environment in different ways, and in many cases the community of core species may be more relevant for testing theoretical predictions. For example, coexistence theory, niche theory, and other related ideas in ecology are largely predicated upon the occurrence of species that are suited to and influenced by their environments, successfully utilizing those environments for food and reproduction (*Umaña et al., 2017*). Analyses carried out in communities that support low proportions of core species may poorly align with ecological predictions that are less applicable to transient species. Indeed, previous work has already shown that a wide range of ecological patterns (e.g., species–area relationships, species abundance distributions) differ depending on whether the analysis focuses on core species, transient species, or the entire community (*Magurran & Henderson, 2003*; *Taylor et al., 2018*). The proportions of core and transient species also vary geographically and therefore influence spatial patterns including species richness gradients (*Coyle, Hurlbert & White, 2013*). Developing general principles regarding the factors that influence the proportion of core species in an assemblage would enable researchers to more effectively compare results between studies and better assess generalities in community ecology.

The extent to which any species is a core, regularly occurring member of an assemblage should depend on the spatial scale over which that assemblage is sampled (Fig. 1A). Consider two extremes: at the scale of 1 m², no bird species would maintain a viable population and be observed in every sampling period. At the scale of the entire North American continent, nearly all species would be annually present at least somewhere within that extent. Thus, the proportion of core species in an assemblage must increase with scale, but the shape of this relationship is less obvious. We expect the shape of the scaling relationship to be a positive decelerating curve (Fig. 1C) because as the extent of a region increases, species that are transient at a local scale will shift to become core species, and the proportion of core species will eventually level off at or below 1 as nearly every member of the regional species pool will have at least one persistent population. This increase will be moderated to some extent by the inclusion of additional transient species from outside the larger regional spatial extent.

Another factor that likely impacts the proportion of core species and the shape of the scaling relationship is environmental heterogeneity, which increases the proportion of transient species likely to occur in an assemblage at a given scale via mass effects (Fig. 1B; *Coyle, Hurlbert & White, 2013*; *Taylor et al., 2018*). Mass effects are more likely in heterogeneous landscapes—that is, when surrounding areas differ in habitat from the focal assemblage–as species poorly adapted to the local environment arrive via dispersal from adjacent source habitats to which they are better suited (*Shmida & Wilson, 1985*).

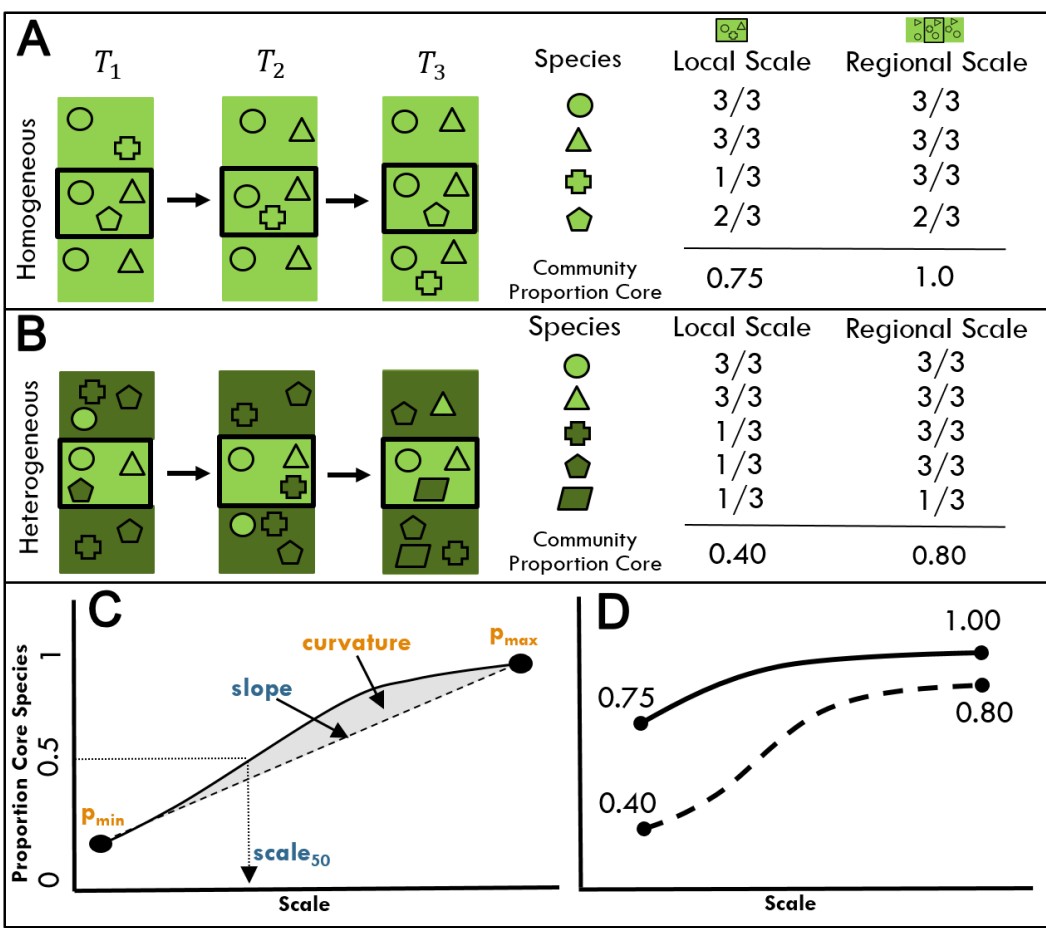

**Figure 1** **The proportion of core species in a community is expected to vary with scale and environmental heterogeneity.** (A, B) Species (symbols) are distributed across an environmentally homogeneous (A) or heterogeneous (B) landscape over three time periods (T1, T2, T3). The temporal occupancy of each species as well as the proportion of core species in the assemblage that occur in 2/3 or more time periods is assessed at both the local (central black boxes) and regional (rectangles) scales. The color of species symbols indicates habitat affinities for landscapes of the same color. (C) A generalized scaling relationship for the proportion of core species in a community. We consider the following parameters from this curve: (1) $p_{min}$, proportion of core species at the minimum spatial scale, (2) $scale_{50}$, the spatial scale at which the community first exceeds 50% core species, (3) $p_{max}$, proportion of core species at the maximum spatial scale, (4) slope, the slope of the line linking the minimum and maximum values, and (5) curvature, calculated as the area between the scaling curve and the straight line connecting min and max values. Parameters in yellow are expected to be negatively related to environmental heterogeneity, while parameters in blue are expected to be positively related to environmental heterogeneity. (D) The proportion of core species in (A) and (B) at local versus regional scales for landscapes of high and low environmental heterogeneity.

Environmental heterogeneity may also constrain habitat availability via the partitioning of space by multiple habitat types within the area delimited by the focal assemblage, and the reduction of area per habitat type relative to environmentally homogeneous sites (*Allouche et al., 2012*). Resources within each habitat may occur at levels below the threshold needed to sustain viable populations (*Allouche et al., 2012*), constraining the proportion of core

species for fine scale sites compared to a homogeneous habitat of the same size. Both effects of environmental heterogeneity on the proportion of core species in an assemblage are expected to be strongest at smaller spatial scales (Fig. 1D). At regional scales, most habitat types will have sufficient resources to sustain viable regional populations and an overall larger proportion of core species. Regardless of the specific mechanism, resource-area tradeoffs or mass effects, we expect heterogeneity will contribute to differences in the shape of the overall relationship between the proportion of core species in an assemblage and spatial scale. While we generally expect this relationship to be positive decelerating as described above, effectively smaller habitat patches in heterogeneous environments may result in the proportion of core species increasing slowly at small scales (Fig. 1D). While determining the specific mechanisms of heterogeneity influencing assemblages is beyond the scope of this paper, verifying a connection between heterogeneity and community assembly is a critical first step.

Here, we make use of a geographically extensive dataset on bird distribution over time which allows us to investigate temporal occupancy, and hence the proportion of core species in an assemblage, over a wide range of spatial scales and environmental contexts. Specifically, we seek to (1) describe the distribution of species' temporal occupancy in ecological assemblages across a gradient of spatial scales, (2) evaluate the relationship between the proportion of core species in a community and the spatial scale at which that community is characterized, and (3) test whether environmental heterogeneity influences that scaling relationship.

## MATERIALS & METHODS

### Bird data

We used data on the distribution of diurnal land birds (excluding raptors) over time from the North American Breeding Bird Survey (BBS), maintained by the United States Geological Survey (*Pardieck et al., 2018*). Our data encompassed the 968 BBS routes across the North American continent that were surveyed continuously over the 15 year period from 2000–2014 that had at least 65 neighboring routes within 1,000 km. Each BBS route is a 40 km roadside transect encompassing fifty 3-minute point count stops, each separated by 0.8 km, in which a single observer records all birds detected within 0.4 km. BBS routes were surveyed each year during the breeding season, typically in June.

Temporal occupancy, the proportion of years a species was observed over some spatially defined area, was calculated for each species at each site at a range of spatial scales (Fig. 2, Fig. S1). We defined the proportion of core species in each assemblage as the proportion of species with temporal occupancy greater than two-thirds (i.e., occurring in at least 11 out of the 15 survey years) following *Coyle, Hurlbert & White (2013)*. We also considered alternative thresholds of temporal occupancy for defining core species (0.5 and 0.75) that produced qualitatively similar results (Figs. S2, S3). Below the scale of a single BBS route, each route was split into non-overlapping segments of 5, 10 or 25 point count stops (Fig. 2), and the proportion of core species was calculated at each spatial scale. To examine spatial scales greater than a single BBS route, for each focal route we sequentially aggregated survey
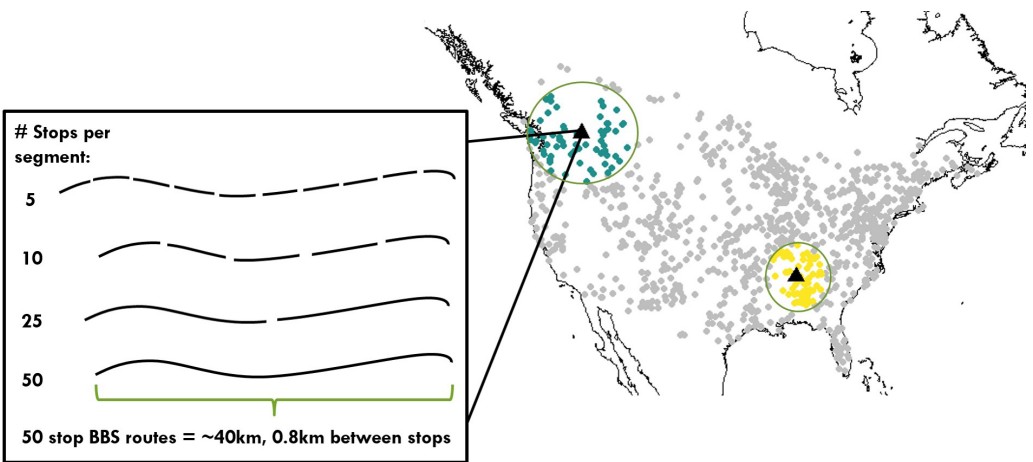

**Figure 2  Distribution of bird communities and range of spatial scales examined for calculating temporal occupancy and the proportion of core species.** Map of North America shows the 968 Breeding Bird Survey routes used in this study, including two examples of the maximum scale examined: 66 survey routes aggregated together, which span variable extents depending on route density. The inset shows a single survey route made up of 50 point count stops, and the spatial scales examined below the level of a route.

data from an increasing number of nearest neighbor routes, up to a maximum regional scale of the focal route together with its 65 nearest neighbors (Fig. 2).

Our regional scale of 66 neighboring routes was chosen because it was the number of neighbors that fell within a radius of 1,000 km of each focal route even in regions of lower route density in the western US (Fig. 2). The entire range of spatial scales we investigated varied from 2.5 km$^2$ for a set of 5 point count stops up to 1,659 km$^2$ for an area of 66 adjacent BBS routes. Because BBS route density varies across the continent, the spatial extent spanned by the 65 nearest neighbors did vary (Fig. 2). However, despite this variation in spatial extent, the total surveyed area characterizing an assemblage was constant (1,659 km$^2$), and this was the aspect of scale we viewed as most critical for our comparisons. While regions of the same sampled area but spanning larger extents may encompass a greater range of environmental variation all else equal, we measured this variation directly (see Environmental Data below).

In addition to spatial scale, we used the total number of individuals observed in the assemblage (community size) as an alternative measure of scale. Community size was found to be a potentially more generalizable measure of scale than area, especially for comparing between taxonomic groups with very different area requirements (*Taylor et al., 2018*).

## Scaling metrics

We derived a series of metrics characterizing the shape of the relationship between the proportion of core species present and scale for each focal route (Fig. 1C). We identified the proportion of core species at the smallest scale ($p_{min}$) and the proportion at the largest scale ($p_{max}$) for each focal route. We also identified the slope of the line linking $p_{min}$ and $p_{max}$ for each focal route, with differences in the slope from focal route to focal route indicative of differences in the increase of the proportion of core species over area. We identified the

scale at which the proportion of core species in the community surpassed the threshold of 0.5 for each focal route ($scale_{50}$). Finally, we characterized the degree of curvature in the relationship between the proportion of core species in the community and scale. As a measure of curvature, we estimated the area between the observed scaling curve and the straight line linking $p_{min}$ and $p_{max}$ by summing the differences between the observed values and the values expected from the linear relationship across all scales (Fig. 1C). If the observed relationship lays along a straight line, this curvature metric would be 0. Positive values indicate positive decelerating relationships in which the observed proportion of core species lies primarily above this straight line (e.g., Fig. 1C), while negative values indicate positive accelerating relationships with observed the proportion of core species lying primarily below this line.

### Environmental data

We acquired raster layers for 0.25 km resolution elevation from Worldclim (*Fick & Hijmans, 2017*), and 0.25 km resolution Normalized Difference Vegetation Indices (NDVI) from the NASA GIMMS group (*Didan, 2015*), and calculated mean NDVI and mean elevation for each focal route within a 40 km buffer of the route's starting coordinates. For each environmental variable, we defined regional heterogeneity around each focal route as the variance in mean values across the set of 65 nearest neighbor BBS routes plus the focal route. In order to assess whether the importance of environmental heterogeneity varied with the spatial scale over which heterogeneity was measured, we also calculated environmental heterogeneity at different scales (from three to 66 neighboring routes). We then examined the Pearson's correlation across all 968 focal routes between heterogeneity and the five scaling metrics describing how the proportion of core species varies across the full range of spatial scales.

## RESULTS

At the scale of a single route (~25 km$^2$), temporal occupancy was bimodal as expected (Fig. 3, dashed line). At larger spatial scales, assemblages were marked by a greater proportion of core species with high temporal occupancy, while at smaller scales, assemblages were characterized by a greater number of transient species and very few core species (Fig. 3). The proportion of core species in a community increased on average in a positive decelerating manner with both measures of spatial scale, although there was substantial variability from route to route (Fig. 4A). At the largest spatial scales, the proportion of core species exhibited reduced variation, with a mean of 83% and ranging from 75%–90%, while at the smallest spatial scales (2.5 km$^2$) the proportion of core species varied from 11–37%. Using community size in lieu of spatial scale greatly reduced this variation in the proportion of core species at the smallest scale (Fig. 4B).

Heterogeneity in elevation and heterogeneity in NDVI had qualitatively similar effects on the overall shape of the relationship between the proportion of core species and spatial scale, although the effects of elevation were stronger for some measures such as *curvature* and *scale$_{50}$* (Fig. 5). Environmentally heterogeneous regions had assemblages with a low proportion of core species at both the smallest ($p_{min}$) and largest scales ($p_{max}$),

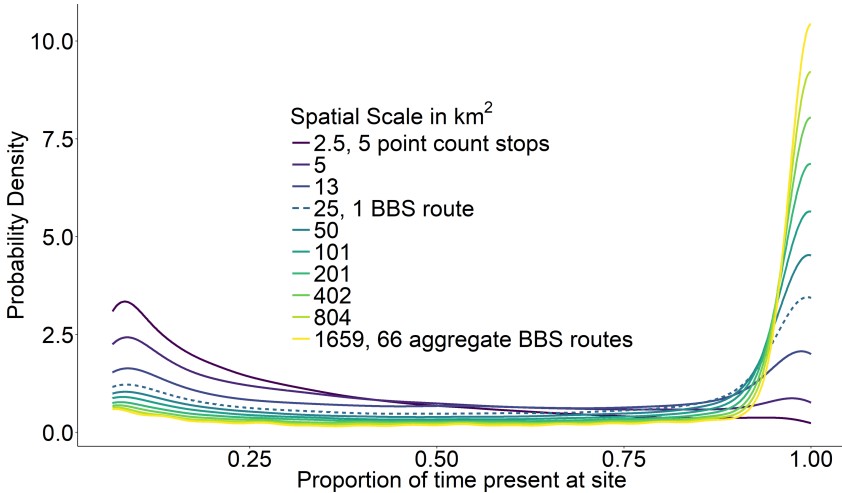

**Figure 3  Average probability densities of temporal occupancy for the bird species present at a site.**
Average probability densities of temporal occupancy for the bird species present at a site, calculated over
ten spatial scales from small (dark) to large (light). Each curve represents the average probability density
across 968 BBS routes at a particular scale. BBS route scale highlighted with dashed line.

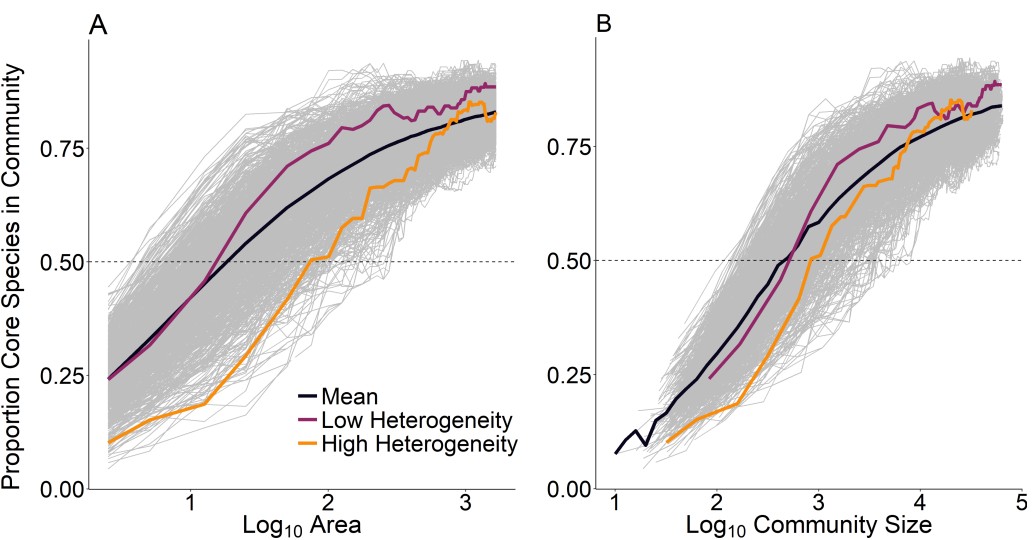

**Figure 4  Proportion of core species present in assemblages as a function of (A) scale as measured by
area and (B) scale as measured by community size.** Each line represents a single focal BBS route; we ex-
amined 968 routes total. Average across all BBS routes indicated by the bold black line. Highlighted routes
exemplify low environmental heterogeneity (purple, Illinois, route 54) and high environmental hetero-
geneity (orange, Utah, route 169).

and communities that experienced the greatest increase in the proportion of core species
between the smallest and largest scales (*slope*). Assemblages in more heterogeneous regions
additionally displayed less positive *curvature* values and a larger spatial scale at which the
majority of species were identified as core (*scale₅₀*).
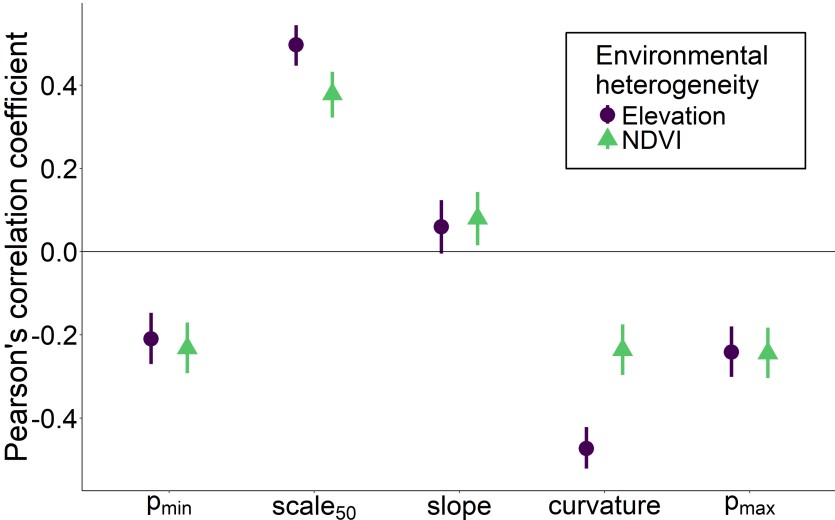

**Figure 5** **Correlation between two measures of regional environmental heterogeneity and five parameters describing how the proportion of core species increases with scale.** Correlation between two measures of regional environmental heterogeneity and five parameters describing how the proportion of core species increases with scale. Bars indicate 95% confidence intervals.

The scale at which environmental heterogeneity was measured also affected the strength of the correlation between heterogeneity and scaling curve metrics (Fig. 6). Specifically, the correlation between all five of the scaling metrics and heterogeneity in elevation was strongest when that elevational heterogeneity was measured at the largest spatial scale, especially for *curvature*, $scale_{50}$, and $p_{max}$. In contrast, heterogeneity in NDVI exhibited the strongest correlations with $p_{min}$, $scale_{50}$, and *slope* parameters when that heterogeneity was measured at scales between 15–25 BBS routes (400–600 km$^2$; Fig. 6). With the exception of *curvature*, heterogeneity in NDVI was a stronger correlate of our scaling metrics than heterogeneity in elevation at these intermediate scales.

## DISCUSSION

Ecologists frequently test hypotheses regarding community assembly and species richness using surveys that reflect a snapshot of a community at a particular point in time. However, it is increasingly recognized that such a snapshot approach fails to differentiate core species from transient species, the former maintaining viable populations and interacting more strongly with their biotic and abiotic environment, and the latter being irregular visitors that are presumably better adapted to other conditions (*Magurran & Henderson, 2003*; *White & Hurlbert, 2010*; *Umaña et al., 2017*). We used a continent-wide dataset on bird assemblages over time to evaluate how the proportion of core species in these assemblages increases with scale and decreases with environmental heterogeneity. Consistent with *Coyle, Hurlbert & White (2013)*, the distribution of temporal occupancy was strongly bimodal at the scale of a single BBS route, reflecting these two distinct groups. However, at scales below the size of a BBS route (<25 km$^2$) few species were present consistently over time, while at

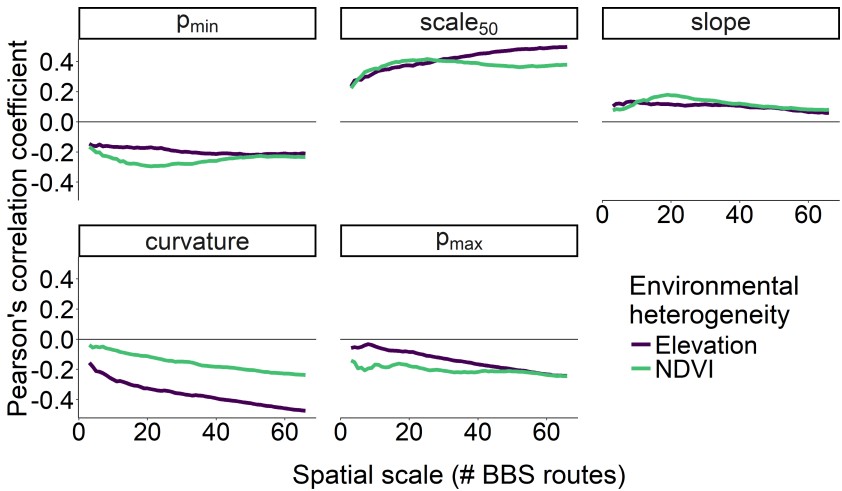

**Figure 6  Correlation between two measures of environmental heterogeneity and five parameters as a function of spatial scale.** Correlation between two measures of environmental heterogeneity and five parameters describing how the proportion of core species increases with scale as a function of the spatial scale over which environmental heterogeneity was characterized.

scales larger than two aggregated BBS routes ($>50$ km$^2$) most species occurred regularly. The smallest scale assemblages ($2.51$ km$^2$) exhibited a fairly wide range in the proportion of core species present (11–37%), at least in part because different sites differed in the overall number of individuals supported. At the largest spatial scales ($1,659$ km$^2$ of surveyed area distributed across a $1,000$ km radius region) there was less variation in the proportion of core species (75–90%). On average, the proportion of core species in a community increased in a positive decelerating manner as a function of spatial scale. As scale increased, so did the probability of including suitable habitat in sufficient quantities to support persistent populations, and species identified as transient at small scales subsequently became core species at larger scales. The exact scale at which a species was observed to become a core species varied with a species' space requirements. For example, a raptor (red-tailed hawk) that occurred at low density over large areas might not be inferred to be a core species until the surveyed area exceeded $100$ km$^2$, while a small songbird (chipping sparrow) might be observed consistently over time at scales as small as $3$ km$^2$ (Fig. S1). Nevertheless, even at the largest scales considered here transient species made up 10% or more of the species observed, presumably representing species that were more adapted to ecozones outside the focal region.

Much of the variation in the shape of the relationship between the proportion of core species in a community and spatial scale can be explained by the regional environmental heterogeneity surrounding the assemblage. Specifically, landscapes with high environmental heterogeneity have proportionally fewer core species, and this effect is strongest at the smallest spatial scales. Consistent with previous findings, we found that environmental heterogeneity was negatively correlated with the proportion of core species (*Coyle, Hurlbert & White, 2013*; *Taylor et al., 2018*). This was true whether characterizing

heterogeneity based on regional variation in elevation or NDVI, but the effect of elevation was as strong or stronger than NDVI at the regional scale (Fig. 6). This is likely because variation in elevation encompasses habitat diversity due to the inclusion of different zones of elevation in addition to differences in slope, hydrology, and other topographic features. Variation in NDVI also presumably captures many of these differences, but perhaps less directly as the habitat variation within a given range of NDVI may not be well captured. Ultimately, regional heterogeneity increases the relative proportion of transient species at local scales via the increased likelihood of mass effects by species better adapted to adjacent habitat types (*Shmida & Wilson, 1985*; *Coyle, Hurlbert & White, 2013*; *Taylor et al., 2018*). Landscapes with low environmental heterogeneity should support communities with low temporal turnover (*Stegen et al., 2013*; *Gaston et al., 2007*), even at small spatial scales nested within the region as these small scale habitats more closely parallel the resources and composition of the region they occur within. Landscapes with a high degree of environmental heterogeneity are more spatially compartmentalized, effectively decreasing the area and resources available per habitat type to support a viable species population (*Allouche et al., 2012*). Thus, in addition to experiencing greater mass effects, any particular habitat type within a heterogeneous region is less likely to encompass sufficient area and resources necessary to sustain viable populations. This logic holds for any set of organisms for which population persistence is a function of environmental suitability. Thus, while birds are highly mobile (and in some cases migratory) relative to many other taxa, the expected effect of heterogeneity on the proportion of core species and its spatial scaling should be qualitatively similar independent of taxon.

These relationships between the proportion of core species and both scale and environmental heterogeneity may help resolve discrepancies between studies regarding the importance of biotic interactions, resource availability, and mass effects for driving community assembly (*Henderson & Magurran, 2014*). Difficulties in synthesizing and generalizing across studies may have arisen from differences in scale and environmental heterogeneity leading to assemblages with different proportions of core species and therefore different apparent mechanisms driving community assembly (e.g., (*Dorazio et al., 2006*; *Emerson & Gillespie, 2008*; *Stein, Gerstner & Kreft, 2014*). For example, competition and environmental filtering have both been proposed to shape community assembly and influence phylogenetic overdispersion and clustering (*Cavender-Bares et al., 2004*; *Mayfield & Levine, 2010*). However, the degree of overdispersion or clustering may also be affected by the proportion of core or transient species in a community. Core species are more likely to compete with each other for resources, and would be expected to contribute the most to overdispersion in competition related traits. In addition, core species are expected to be better suited to the local climate or habitat compared to transient species, and so would be expected to exhibit greater clustering of environmental tolerance traits. At small spatial scales, the proportion of transient species will be higher, resulting in a lower likelihood of discerning a nonrandom assembly pattern. The proportion of core species is lowest at small scales, and yet the processes driving core species assembly, like competition, should be most important at these scales where individuals are more likely to interact (*Allouche et al., 2012*). This may result in seemingly conflicting, or altogether masked, patterns of

community assembly in large meta-analyses that include studies conducted at a wide range of scales from disparate taxonomic groups.

For example, a meta-analysis on phylogenetic clustering versus overdispersion conducted by *Emerson & Gillespie (2008)* found an assemblage of Cuban *Anolis* lizards to exhibit unstructured or seemingly random patterns of phylogenetic assembly while an assemblage of dusky salamanders exhibited strong phylogenetic overdispersion. This result may reflect a lack of generality in the degree of overdispersion in community assembly. However, these two studies were carried out at very different spatial scales. Our results suggest that the *Anolis* study may have found unstructured patterns as a result of the small scale of the study, where we would expect there to be a low relative proportion of core species making it difficult to detect nonrandom patterns. Comparisons of assembly patterns between groups that differ strongly in space use and area requirements (100 m$^2$ means something very different for a bird assemblage compared to a plant assemblage) may further strain effective comparisons. When testing for aspects of community structure, restricting the analysis to core species should increase the power to detect non-random trait assembly patterns and improve the search for generality.

Macroecological analyses of core and transient species use observational time-series to identify these two groups. While this is the only practical way to accomplish this classification at scale (considering thousands of species-site combinations), it can result in two types of classification errors: species may be inferred to be transient when they are actually core (a false negative), and they may be inferred to be core when they are actually transient (a false positive). False negatives lead to underestimates of the proportion of core species, and they are expected to occur primarily at intermediate spatial scales. At small scales, few species actually maintain viable populations and nearly all species are truly transient. At large scales, even species that occur at low density will reliably be observed somewhere from year to year, and so nearly all species are truly core. The fact that false negatives will be most common at intermediate scales implies that the "true" curve scaling the proportion of core species with area or community size has similar $p_{min}$, $p_{max}$, and slope values to the observed curve. However, if the "true" proportion of core species at intermediate scales is actually higher than observed due to these false negatives, then we would expect the scale at which that proportion exceeded 0.5 ($scale_{50}$) to be slightly smaller and the estimates of curvature to be slightly larger than observed. False positives are expected to occur primarily at small scales in regions of high environmental heterogeneity. A species that does not sustain a viable population at a local sink site but does in the surrounding region may appear to be a core species at that sink site because neighboring sites support sufficient populations to ensure regular immigration to the sink site. However, the fact that environmental heterogeneity had a negative effect on the observed proportion of core species implies that this bias is minimal. Future research using simulation models to assess misclassification rates for communities across different scales and levels of environmental heterogeneity, and for species with different densities and detection rates, will be necessary for evaluating the extent to which spatial scales and heterogeneity influence classification errors. Alternatively, using stricter thresholds of temporal occupancy for determining the proportion of core species may help reduce the likelihood of false positives (Figs. S2, S3).

## CONCLUSIONS

The distinction between core and transient species is increasingly recognized as being important for properly testing predictions and comparing ecological systems (*Magurran & Henderson, 2003*; *Coyle, Hurlbert & White, 2013*; *Supp, Koons & Ernest, 2015*; *Umaña et al., 2017*; *Taylor et al., 2018*), making it critical to understand the factors that influence the relative proportion of these two different groups. Here, we have shown that the proportion of core species in an assemblage is positively associated with spatial scale and negatively associated with environmental heterogeneity. The relative proportion of these two groups of species influences a number of essential patterns in community ecology, including the species–area relationship, species-abundance distribution, temporal turnover, and geographic patterns of biodiversity (*Magurran & Henderson, 2003*; *Taylor et al., 2018*). All of these patterns are scale-dependent, and investigators have typically assumed an effect of scale itself (*Adler et al., 2005*; *Rahbek, 2005*; *Green & Plotkin, 2007*). Our results suggest an extra layer of complexity in that scale influences the proportion of core and transient species which may influence ecological patterns independent of scale. Future work attempting to understand the different ways in which scale influences ecological systems should consider this indirect influence of scale via the proportion of core members of an assemblage. In general, an understanding of the factors that influence the prevalence of core species is critical for the proper interpretation of synthetic meta-analyses and the evaluation of ecological theory.

## ACKNOWLEDGEMENTS

We are grateful to S. Taylor, J. Coyle and two anonymous reviewers for comments that improved an earlier version of this manuscript, and to the thousands of volunteers who annually contribute to data collection for the North American Breeding Bird Survey.

### Funding

This work was made possible by funding from the National Science Foundation through grant DEB-1354563 to Allen H. Hurlbert and Ethan P. White and by the Gordon and Betty Moore Foundation's Data-Driven Discovery Initiative through grant GBMF4563 to Ethan P. White. The funders had no role in study design, data collection and analysis, decision to publish, or preparation of the manuscript.

### Grant Disclosures

The following grant information was disclosed by the authors:
National Science Foundation: DEB-1354563.
Gordon and Betty Moore Foundation's Data-Driven Discovery Initiative: GBMF4563.

### Competing Interests

Ethan P. White is an Academic Editor for PeerJ.

## Author Contributions

- Molly F. Jenkins conceived and designed the experiments, performed the experiments, analyzed the data, contributed reagents/materials/analysis tools, prepared figures and/or tables, authored or reviewed drafts of the paper, approved the final draft.
- Ethan P. White conceived and designed the experiments, contributed reagents/materials/analysis tools, authored or reviewed drafts of the paper, funding, original project proposal.
- Allen H. Hurlbert conceived and designed the experiments, contributed reagents/materials/analysis tools, authored or reviewed drafts of the paper, approved the final draft, funding, original project proposal.

## Data Availability

Master's thesis research: data, methods, results, and figures: https://github.com/mollyfrn/core_scale.

## Supplemental Information

Supplemental information for this article can be found online at http://dx.doi.org/10.7717/peerj.6019#supplemental-information.

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
