# Peer review of "The proportion of core species in a community varies with spatial scale and environmental heterogeneity"

_PeerJ, doi:10.7717/peerj.6019_

## Round 0.1 · original submission · Minor Revisions

In addition to paying close attention to the reviewer's comments, I think that (1) you could explain the metrics a little more clearly for readers, particularly how temporal data were incorporated in their calculation, which wasn't easy to understand and (2) your arguments would benefit from mentioning the ecology of the species in whether they are transient or core at different scales. For birds, this depends on their home range sizes, so a 'misclassification' of a particular species will relate to its ecology as much as to the scale, as discussed.

Reviewer 1 ·

Basic reporting

no comment

Experimental design

no comment

Validity of the findings

no comment

Additional comments

This paper examines the role of spatial scale and environmental heterogeneity on the proportion of core and transient species in bird communities by using an extensive dataset on bird surveys for North America combined with climatic data. The main findings show that the proportion of core species increases with the spatial scale while transient species tend to decrease. In addition, they found that this relationship is affected by habitat heterogeneity, in which the proportion of core species at a given scale is lower in sites with high heterogeneity than in sites with low heterogeneity. The authors discuss the results in terms of the potential implications that the variable proportion of core and transient species has on the inferences of ecological processes.

The manuscript is well written and I found the topic interesting and appealing to a broad scientific audience; the methods are clear, the analyses well performed and the figures relevant. My only concern relays on the interpretation of the results and the role of ecological processes inferred: (1) It has been already stated that patterns of phylogenetic distribution in natural communities are a weak approach for inferring competition or filtering processes. So, I found the discussion to be somehow misleading in this respect (lines 258-259). I believe the authors can address this comment by rephrasing these sentences (2) Authors discuss how the differences in proportions of core and transient species influence the inference of ecological processes. However, they mostly focus on competition. I wonder if they have any thoughts regarding the role of immigration. Immigration rates are dependent on the spatial scale, being dominant in large areas and at the same time, immigration should be more important for transient than for core species.

·

Basic reporting

The reporting of the manuscript is generally clear and professional; the authors comply with the requirements of PeerJ. Nonetheless, some issues remain. For instance, specific measurements, especially in the methods could be explained in more detail and more intuitively (less technically and explaining the implications of, e.g., metrics, see methods section). The authors come up, from time to time, with concepts or definitions not previously introduced; this may prove confusing to the readership (please see comments in the PDF). The structure and background are sufficient, and the core message is relevant to the hypothesis.

Experimental design

While the authors describe in detail the data and the scales (except few details, please see comments in the PDF), several sections in the analysis need further elaboration for the paper to be reproducible (please see comments in PDF). There is no acknowledgment of the possibility and implications of overlapping of species or repeated occurrences (because of, e.g., migration or possibly overlapping of sampling areas) when pooling together, e.g., the BBS routes.

Validity of the findings

Regardless of the importance of the subject for the understanding of ecological processes, the authors need to deepen the explanation of their analysis and discuss the several particularities of their system of study. Whether the authors wish to generalize their findings or not the fact remains that they are working with birds which have a particular behavior, they are mobile, and they migrate. Furthermore, the results seem to be interpreted subjectively without rigorous tests supporting what the authors indicate, thus making interpretation subjective. In my opinion that is the most pressing issue I have encountered in the manuscript (please see my comments in the PDF). Finally, the application and importance of the results to current ecological theory do not seem apparent from the discussion.

Additional comments

The authors describe the variation of the proportion of core and transient species in bird communities along different spatial scales. They also assess the effect of environmental heterogeneity on this scaling relationship.
Abstract: the authors sustain that scale influences the proportion of core species. It is clear that there is an effect of both scale and environmental heterogeneity, but it is tough to discern to which degree the proportion of core species is a result of area size alone, size and heterogeneity combined or heterogeneity alone (this is mentioned in the discussion without real support from the results).
In the introduction the authors document and justify well the core-transient distinction, providing enough context for their work. Although, they raise very high expectations by encouraging the development of general principles concerning factors influencing the proportion of core species. The implication that their work may lead to this is counterintuitive because although the analysis does cover a considerable spatial extension, it focuses on a particular system (birds). Birds are mobile and migrate; these are particularities have not been mentioned nor discussed. Also, the implication that this work may be applied in a general context is not supported in the discussion. If this was the intention of the authors, they could have come up with recommendations on how to apply their analysis or findings to other systems. Also, the authors could compare theirs to findings from different systems.
The authors do not discuss their findings thoroughly. Instead, the discussion also contains some more arguments about the validity of differentiating core and transient species. E.g., How can the influence of scale and environmental heterogeneity in the proportion of core species help dissolve discrepancies in these ecological studies? Examples or suggestions of the sort are missing in the discussion.
The figures (5 and 6) could include a very brief description of the metrics.

---

## Round 0.2 · accepted · Accept

Thank you for your comprehensive responses to the reviewer comments. One final change is to remove capitals from some bird names that were added in the revision (lines 247-250) - please do not capitalise common names (unless they contain a proper noun).

#